# Glow-in-the-Dark Patterned PET Nonwoven Using Air-Atmospheric Plasma Treatment and Vitamin B2-Derivative (FMN)

**DOI:** 10.3390/s20236816

**Published:** 2020-11-28

**Authors:** Sweta Narayanan Iyer, Nemeshwaree Behary, Vincent Nierstrasz, Jinping Guan

**Affiliations:** 1ENSAIT-GEMTEX, F-59100 Roubaix, France; nmassika.behary@ensait.fr; 2Université Lille Nord de France, F-59000 Lille, France; 3Textile Materials Technology, Department of Textile Technology, Faculty of Textiles, Engineering and Business, University of Borås, SE-50190 Borås, Sweden; Vincent.Nierstrasz@hb.se; 4College of Textile and Clothing Engineering, Soochow University, Suzhou 215021, China; guanjinping@suda.edu.cn

**Keywords:** flavin mononucleotide (FMN), glow-in-the-dark, yellow-green fluorescence, biobased, PET nonwoven textile

## Abstract

Flavin mononucleotide (FMN) derived from Vitamin B2, a bio-based fluorescent water-soluble molecule with visible yellow-green fluorescence, has been used in the scope of producing photoluminescent and glow-in-the-dark patterned polyester (PET) nonwoven panels. Since the FMN molecule cannot diffuse inside the PET fiber, screen printing, coating, and padding methods were used in an attempt to immobilize FMN molecules at the PET fiber surface of a nonwoven, using various biopolymers such as gelatin and sodium alginate as well as a water-based commercial polyacrylate. In parallel, air atmospheric plasma activation of PET nonwoven was carried for improved spreading and adhesion of FMN bearing biopolymer/polymer mixture. Effectively, the plasma treatment yielded a more hydrophilic PET nonwoven, reduction in wettability, and surface roughness of the plasma treated fiber with reduced water contact angle and increased capillary uptake were observed. The standard techniques of morphological properties were explored by a scanning electron microscope (SEM) and atomic force microscopy (AFM). Films combining each biopolymer and FMN were formed on PS (polystyrene) Petri-dishes. However, only the gelatin and polyacrylate allowed the yellow-green fluorescence of FMN molecule to be maintained on the film and PET fabric (seen under ultraviolet (UV) light). No yellow-green fluorescence of FMN was observed with sodium alginate. Thus, when the plasma-activated PET was coated with the gelatin mixture or polyacrylate bearing FMN, the intense photoluminescent yellow-green glowing polyester nonwoven panel was obtained in the presence of UV light (370 nm). Screen printing of FMN using a gelatin mixture was possible. The biopolymer exhibited appropriate viscosity and rheological behavior, thus creating a glow-in-the-dark pattern on the polyester nonwoven, with the possibility of one expression in daylight and another in darkness (in presence of UV light). A bio-based natural product such as FMN is potentially an interesting photoluminescent molecule with which textile surface pattern designers may create light-emitting textiles and interesting aesthetic expressions.

## 1. Introduction

Luminescent materials have a wide range of applications in different industries, including apparel, furnishings, printing and warning signs, safety alerts, or design features in the interior, architectural, and automotive textiles. Today glowing fabrics are mainly based on electrical wires, optical fibers inserted in the fabric, and connected to a battery to produce light [1,2,3,4,5,6,7]. Designing a two-phase glow-in-the-dark pattern on textiles offers a new design resource for textile surface pattern designers to promote creativity in design. The glow-in-the-dark pattern can be implemented by the printing of photoluminescent pigments or dyes. Photoluminescent pigments absorb natural or artificial light, generally ultraviolet (UV) rays, and gradually emit light. The pigments become highly visible in ambient darkness (dark light, which is blue UV light), creating the effect generally known as “glow-in-the-dark” [8,9,10,11,12,13]. Photoluminescence comprises two light-emitting processes widely known as fluorescence and phosphorescence. Various kinds of organic substance have been identified for their fluorescent ability. In general, the fluorescent dyes absorb radiation at short wavelengths and emit radiation at longer wavelengths. Fluorescent substances have been known for a long time and, currently, different categories of dyes such as xanthene, rhodamine, and pthalein derivatives exhibit a photoluminescent property. Optical brighteners, also known as fluorescence-whitening agents, are mostly introduced to the textiles, usually during finishing, wherein they absorb UV light, and visible light emitting at a different wavelength provides a brighter appearance on the fabric. However, the washing of the optical brighteners during the laundry process has led to environmental concerns [14].

Luminous polypropylene composite nonwovens containing rare-earth strontium aluminate and functional additives have also been developed [15]. The photoluminescent molecules or particles used today, such as rare earth-doped strontium aluminate (SrAl_2_O_4_: Eu^2+^, Dy^3+^), are high-cost rare earth luminescent materials and also seem to have a health impact on workers [16,17,18]. Therefore, it is necessary to find alternative bio-based photoluminescent molecules to obtain photoluminescent and glow-in-the-dark textiles. The flavin mononucleotide (FMN, C_17_H_21_N_4_O_9_P) is a biomolecule produced from riboflavin (Vitamin B2) by enzymatic reaction. It is a phosphoric ester of riboflavin that naturally occurs in milk, green leafy vegetables, and also in plant and animal cells. FMN is present in bound forms naturally in foods. It is composed of a triheterocyclic group called isoalloxazine, and a lateral chain of phosphate group allows the molecule to be soluble in water compared to riboflavin. The isoalloxazine group gives these molecules a visible color (yellow in the case of FMN), along with fluorescent properties [19]. In this study, FMN as a fluorescent dye on textiles has been explored to discover the new scope for photoluminescent and glow-in-the-dark textiles, like those produced by Kooroshnia [8], but using a biobased resource. Indeed, FMN exhibits fluorescence that can be observed in the dark in the presence of UV-light.

Therefore, the main aim of this paper is to study the application of flavin mononucleotide (FMN) to produce glow-in-the-dark patterns on polyester nonwoven, with the possibility of one expression in daylight and another in darkness. Our previous study [20] demonstrated that FMN could diffuse inside cellulosic fiber textiles and produce photoluminescent textile materials. However, the FMN could not dye the polyester nonwovens, thus, in this paper, the combined use of FMN bearing biopolymers/acrylate-based polymers has been investigated using coating and padding with prior surface activation of nonwoven polyester (PET) to produce either uniform luminescent PET nonwoven panels or glow-in-the-dark patterns using printing. In the first part of the study, air atmospheric plasma activation of PET nonwoven was carried to produce hydrophilic PET nonwoven. The surface changes after plasma treatment were characterized by water contact angle (wettability/capillarity) and atomic force microscopy (AFM) measurements along with morphological analysis using a scanning electron microscope (SEM). Two biopolymers (gelatin and alginate) and a commercial polyacrylate were tested. Fluorescence/photoluminescence of the films and the plasma nonwoven PET coated/padded or printed with biopolymer/polymer and FMN mixture were then visualized under UV light and evaluated using photoluminescence spectroscopy.

## 2. Materials and Methods

### 2.1. Materials

#### 2.1.1. Polyester (PET) Nonwoven

A 100% PET nonwoven structure manufactured at the European nonwoven platform center in Tourcoing, France, was used for the experimental study. It was formed by needle punch combined with the hydroentanglement process. The nonwoven possessed 950 µm thickness, 93% porosity, 230 g/m^2^ areal density, and 12 µm average diameter. The nonwoven was cleaned to be free from surface impurities and spinning oil, and the sample cleanliness was confirmed by measuring the surface tension of rinse water. It remained constant at 72.6 mN/m, which is equivalent to the surface tension of pure water.

#### 2.1.2. Chemicals

Flavin mononucleotide (FMN) or riboflavin 5′-monophosphate sodium salt hydrate, gelatin from porcine skin, sodium alginate, sodium phosphate buffer was purchased from Sigma Aldrich. Also, a water-based commercial acrylate ‘Appretan N 9211’ was procured from Archroma. The chemical details are elaborated below (Table 1) as per the supplier specifications and the literature study.

### 2.2. Methods for Nonwoven Treatment and Application

#### 2.2.1. Plasma Treatment

A 100% nonwoven PET was plasma treated with air atmospheric plasma (ATMP). A square size 50 × 50 cm of nonwoven PET was cut as per the plasma machine’s electrode length. The treatment was performed on a Coating Star plasma treatment set up manufactured by Ahlbrandt system (Germany). The sample was placed in between the two electrodes by maintaining the following machine parameters; the electrical power of 1 kW, frequency of 26 kHz, speed of the textile treatment 2 m/min, electrode length of 0.5 m, and an inter electrode distance of 1.5 mm along with treatment power of 60 kJ/m^2^. The nonwoven was treated on both sides, and the plasma-treated samples were then kept away from light to prevent them from aging [22].

#### 2.2.2. Preparation of Biopolymer/Flavin Mononucleotide (FMN) Film

Gelatin and alginate gels of 5% concentration were prepared, and the viscosities were measured using the rheoplus equipment at 20 °C.

Based on the respective biopolymer’s gelatinization temperature, as mentioned in Table 2, the gel was prepared using distilled water. The gels were then introduced with 1 mL of 1% and 5% FMN solution mixing it homogeneously on the magnetic stirrer for 5–10 min. Using the homogeneously mixed biopolymer and FMN mixture, films were formed in polystyrene (PS) Petri dish.

#### 2.2.3. Application of Biopolymer/Acrylate FMN Mixture on Plasma-Activated PET Using Screen Printing, Coating and Padding

The biopolymer/FMN mixture was applied onto plasma treated nonwoven using the screen printing/coating technique, as shown in Figure 1. The biopolymer/FMN mixture was screen printed on plasma-treated nonwoven using the laboratory’s flat bed screen printer (Figure 1a). The printing formulation was spread across the screen frame and squeezed using a squeegee to pass through the screen onto the textile substrate. Furthermore, a metal rod was placed over the fixed screen and continuously coated onto the textile substrate (Figure 1b). A padding method was used for PET nonwoven fabric coloration using photoluminescent FMN with acrylate binder. A laboratory scaled padder (Werner Mathis AG, Niederhasli, Switzerland) was used for padding. The pressure was set to 2 bar with a rotation speed of 2.5 m/min. The typical pad coating arrangement can be seen in Figure 1c. This method allowed coating, and the polymer/FMN mixture could penetrate between the fibers, as the textile was initially dipped, and the rollers then squeezed out the excess mixture. All the samples were dried at room temperature in a dryer overnight and also washed with water at 30 °C (4 times), followed by rinsing in cold water to study the preliminary wash fastness.

### 2.3. Characterization Methods of Treated Nonwoven

#### 2.3.1. Capillary Measurement and Water Contact Angle

A wicking test using a tensiometer from GBX Instrument (France) was performed to calculate the water contact angle and the capillary uptake of the nonwoven PET, as described in our previous paper [23]. A rectangular sample of the nonwoven was connected to the tensiometer at the weighing position. It was progressively brought into contact with the surface of water placed in a container. On immediate contact with the water surface, a sudden increase in weight (Wm) was measured due to meniscus formation on the fabric surface, and the water contact angle was then calculated using Equation (1).
(1)$$W_{m}\cdot g=\gamma_{L}\cdot \cos\theta \cdot p$$

#### 2.3.2. Atomic Force Microscopy (AFM) Analysis

The Dimension Icon instrument AFM, Scanasyst, Veeco from Bruker was operated in contact mode was used to investigate the surface roughness before and after the plasma treatment. The working principle of the equipment is elaborated in the literature [24].

#### 2.3.3. Scanning Electron Microscope (SEM) Analysis

Before and after plasma treatment, the morphological property of nonwoven PET was investigated using a Hitachi S 4800 cold field-emission scanning electron microscope (SEM). Before SEM analysis, each textile of 5 × 5 mm as sputtered with gold and placed in the electron microscope stage with conductive adhesive.

#### 2.3.4. Rheology

The biopolymer/FMN mixture’s rheological property was measured using a modular compact Anton Paar rheometer (Physica MCR500, Austria). The viscosity was recorded at the highest measurable shear rate of the instrument, 10,000 s^−1^, with a temperature range from 20 to 60 °C.

#### 2.3.5. Visual Observation under Daylight and Ultraviolet (UV) Blue Light

The FMN printed or coated fabrics were exposed to a monochromatic UV lamp of 370 nm using the set up described previously [20]. The light emitted by the fabric in the form of fluorescence was captured using a camera. All the samples were also observed under daylight, and respective images were captured.

#### 2.3.6. Absorbance Spectrophotometry and color strength (K/S) determination

The aqueous solution of FMN was analyzed using VWR UV Visible spectrophotometer (Model No. UV-3100 PC). A Konica-Minolta CM3610 spectrophotometer was used for color strength evaluation on printed/coated fabrics. The measurements were evaluated in terms of its color coordinates values, which are L* (lightness), a* (redness-greenness), and b* (yellowness-blueness), and the reflectance values measured for wavelength ranging between 360 to 700 nm provided the K/S values.

#### 2.3.7. Photoluminescence Measurement

The photoluminescence spectra of FMN-printed/coated fabrics were acquired using a triple monochromator Dilor RT 30 and then detected using a Hamamatsu R943 photomultiplier in the photon-counting mode with incident power of 3 mW. The spectra were recorded at room temperature and scanned using fixed illumination at wavelengths 364 nm and 470 nm line of a 30 mW argon laser with a 100 mm focal length lens, producing a 30 micron diameter spot on the sample.

## 3. Results

### 3.1. Characterization of the Plasma-Activated PET nonwoven

The capillary uptake of untreated nonwoven PET was observed to be null, which then increased up to 1460 mg for the plasma-treated PET nonwoven (Table 3). Thus, the increase in capillary uptake and reduction in water contact angle to 0° as compared to 141° of the untreated PET nonwoven reveals the increase in surface energy of plasma treated PET fiber. As confirmed in our previous studies [25], the hydrophilic terminal carboxyl and hydroxyl functional groups were introduced on the fiber surface. The SEM images of untreated PET showed a relatively smooth PET fiber surface (Figure 2a). However, the plasma-treatment PET fiber surface showed little surface etching on the cylindrical PET fiber, as shown in Figure 2b.

Figure 3 shows the AFM topographical images (1 µm × 1 µm) of PET fiber surface before and after the plasma treatments. It is seen from Figure 3a that the untreated PET surface shows a smoother surface and the plasma-treated surface revealed a scale-like surface structure with increased roughness, as shown in Figure 3b.

### 3.2. Rheological Properties

Mixtures of alginate and gelatin with FMN were studied by rheology at two different temperatures from 20 to 60 °C at varying shear rates of 100, 200, 300, 400, and 500 s^−1^. The rheological properties of the printing paste play one of the essential roles in textile printing. The analysis revealed that gelatin and alginate mixture [21,26], display non-Newtonian pseudoplastic behavior [27], thus facilitating the coating and the printing process (Figure 4 and Figure 5).

### 3.3. Visual Characterization

#### 3.3.1. Observation of Biopolymer Film with FMN

The biopolymer film formed with FMN was observed under daylight and UV chamber to see the fluorescence effect of FMN 5% of alginate and gelatin with 1 mL and 5 mL of 5% FMN, respectively.

In the absence of FMN, alginate, gelatin formed transparent films when the corresponding mixtures were dried at room temperature in different Petri dishes. These films did not adhere to the hydrophobic polystyrene Petri dish. Furthermore, the FMN was introduced in each biopolymer/polymer mixture, gelatin and alginate films formed each a yellowish-brown layer, as shown in Table 4. However, yellow-green fluorescence was only observed for gelatin films under UV light (370 nm).

#### 3.3.2. Visual Characterization of PET Nonwoven Coated and Printed with FMN, under Daylight and UV Light, Compared to PET Subjected to Dyeing with FMN

Table 5 shows PET fabric images before and after being subjected to dyeing procedure using FMN solution at 130 °C, using the general dyeing method as described in our previous work [20]. The cellulosic fabrics were colored yellowish-brown in daylight and yellowish-green under UV light. However, no such phenomenon was observed for the dyed polyester-PET fabric, which had an inherent pale blue fluorescence under UV light. Indeed the FMN being water-soluble does not diffuse effectively inside the hydrophobic PET fiber.

Thus, the coating of PET nonwoven was done using an FMN and gelatin mixture. Table 6 shows the FMN fluorescence effect on the PET-coated textile sample.

#### 3.3.3. Visual Characterization of PET Nonwoven Printed with Gelatin/FMN

##### Screen Printing

Yellow green fluorescence at 570 nm was due to fluorescence of FMN in gelatin film immobilized on the PET nonwoven, and blue fluorescence as shown in Table 7 was inherent to the polyester nonwoven under blue UV light (photoluminescence of 16,000 a.u. at 443 nm for excitation wavelength at 364 nm).

### 3.4. UV–Visible Spectroscopy Analysis of FMN Solution

As depicted in Figure 6, the UV-visible spectra of 2 × 10^−4^ M FMN solution revealed maximum absorption peaks at 289 nm, 374 nm, and 445 nm, respectively.

### 3.5. K/S and Photoluminescence Intensity Evaluation

#### K/S of Gelatin and FMN Coating on PET Nonwoven before and after Plasma Treatment

A uniform mixture of FMN moiety in gelatin was prepared using 5% gelatin mixed with 5 mL of 5% FMN solution during heating of gelatin solution at 60 °C. This mixture was then coated on untreated and plasma treated PET fabric. The photoluminescence intensity and the color strength of all the treated fabrics were evaluated, as shown in Figure 7.

The characteristic fluorescence property of the FMN was observed on PET-coated textiles. The maximum fluorescence intensity was observed at 530 to 540 nm for FMN coated on untreated PET nonwoven. Furthermore, the intensity maxima were at 570 nm for the FMN coated on plasma treated PET fabric samples. The untreated PET nonwoven coated sample revealed a similar fluorescence maximum wavelength as that observed for the FMN solution. However, a shift of 40 nm wavelength was observed for the plasma PET-coated sample, which might be due to the intensity spectrum and energy difference of the laser beam [28].

In addition, the maxima absorbance for FMN coated on untreated PET nonwoven was observed at 360 and 450 nm with a K/S value around 5. After washing, the K/S value decreased drastically while the photoluminescence intensity was observed to be increased from 5 × 10^3^ a.u. (unwashed) to 1.4 × 10^4^ a.u. (washed). However, the FMN coated on plasma-treated PET nonwoven showed coloration even after washing with K/S values around 8. The fluorescence intensity was also increased after washing from 1.7 × 10^4^ to 2.2 × 10^4^ a.u. The variation in intensity observed was due to the concentration of FMN, which influences the fluorescence intensity where ionic strength log k quenches the fluorescence intensity [29].

The coating application done in plasma-treated nonwoven PET fabric seems to increase the durability. Thus this FMN bearing gelatin mixture can be used as an alternative for printing or coating purposes.

### 3.6. Comparative Analysis of K/S and Photoluminescence Intensity of FMN-Coated and Padded with Gelatin and FMN

As tabulated below (Table 8), for excitation at 470 nm, maximum photoluminescence at 570 nm as high as 30,000 a.u. was recorded for FMN-coated plasma treated PET nonwoven using gelatin after wash.

At this excitation wavelength (470 nm), the uncoated PET plasma treated nonwoven showed a low photoluminescence value of 7000 a.u. However, with excitation wavelength in the UV region (364 nm), the nonwoven PET equally showed intrinsic photoluminescence of 16,000 a.u at 443 nm, which explains the blue fluorescence observed in PET regions where gelatin/FMN were absent.

The maximum photoluminescence intensity of the FMN coated or padded PET nonwovens were observed at 570 nm for both excitation wavelengths at 364 nm and 470 nm. Immediate photoluminescence at a higher wavelength of 570 nm confirmed the fluorescence behavior of the fabric. The photoluminescence intensity before and after washing was studied. The fluorescence intensity remained persistent even after washing due to the plasma treatment and the crosslinking of FMN with the acrylate-based binder.

## 4. Discussion and Conclusions

The printing and pad dyeing of polyester-based fabrics generally involve disperse dyes, which can sublimate readily and are deposited at the surface, subjected to the thermosol process for the diffusion of dyes inside the polyester fiber. The fluorescent FMN molecule’s polar nature does not allow its diffusion inside the hydrophobic polyester fiber, using the dye exhaustion process. Thus, hydrophilic biopolymers such as gelatin and alginate were selected for proper FMN dispersion and use as a thickener in a screen printing process. They can also act as a binder to fix FMN on the PET fiber surface. In order to immobilize the FMN efficiently at the PET fiber surface, the hydrophilic polymers played a key role in binding the molecule. The gelatin and FMN film formed a yellowish-brown layer, which exhibited yellow-green fluorescence under UV light (370 nm). The hydrophilic polymers have no adhesion on the untreated hydrophobic polyester without plasma, and they are, therefore, washed easily. The gelatin/FMN was utterly washed out from the untreated hydrophobic PET. Air atmospheric plasma treatment increased the PET nonwoven’s hydrophilic behavior, with reduced contact angle and increased capillary uptake. Polar functional groups such as carboxylic and hydroxyl are formed due to the scissions of ester bonds by the plasma discharge. This increase hydrophilic behavior allows a more homogeneous fabric and fiber coating (through coating and printing) and improved adhesion of the hydrophilic films onto the PET plasma-activated fiber surface. Polar group creation and the polyester fiber’s increased surface roughness after plasma treatment would explain the increased adhesion of the hydrophilic polymers onto the plasma-treated polyester fibers.

Although different films were formed by drying the FMN-biopolymer (gelatin and sodium alginate) solution in a Petri dish, only gelatin showed the greenish-yellow fluorescence of the FMN. No prominent yellow-green fluorescence was seen with alginate, which could probably be due to the crosslinking behavior and less stable environment in the presence of alginate network than the gelatin network, thus decreasing its fluorescence effect. The use of gelatin as crosslinker and coating applications has been extensively studied due to the viscoelasticity, stability, and biocompatibility [30,31,32]. Thus, the coating of gelatin and FMN mixture on plasma-treated polyester nonwoven allowed to produce an entire panel and create fluorescent patterns of green-yellow fluorescence observed under UV light.

The commercial acrylic binder is a water-based binder used to bind hydrophobic pigments at the textile surface while pigment dyeing through the padding process. After applying the water-based binder with FMN on the PET nonwoven through padding, the binder crosslinking allowed us to immobilize FMN on the polyester nonwoven.

Both gelatin/FMN coating produced glow-in-the-dark panels of the polyester nonwoven. However, with the gelatin, a more rigid photoluminescent panel was obtained, compared to the more flexible panel obtained with the less viscous commercial acrylic resin. The plasma treatment enhanced color strength. After washing, K/S decreases slightly; however, the photoluminescence increases. Photoluminescence is higher for the gelatin/FMN-coated samples.

Hence, it can be concluded that vitamin B2 derivative flavin mononucleotide can exhibit a fluorescence property and yield biobased photoluminescent panels and the glow-in-the-dark pattern on polyester nonwoven, using coating and printing techniques after plasma activation of the PET nonwoven. Gelatin is an ideal polymer coating and binder to observe the fluorescence properties of FMN in printed patterns. Coating application on plasma-treated PET nonwoven results in a photoluminescent polyester nonwoven with improved wash fastness properties. The screen printing technique used for the study allowed us to create a glow-in-the-dark pattern on textiles with the possibility of one expression in daylight and another in darkness. Bio-based natural products such as the FMN are potentially interesting photoluminescent molecules with which textile surface pattern designers may create light-emitting textiles and interesting aesthetic expressions. Further work should focus on the influence of time and light irradiation on the luminescence intensity and coloration.

## Figures and Tables

**Figure 1 sensors-20-06816-f001:**
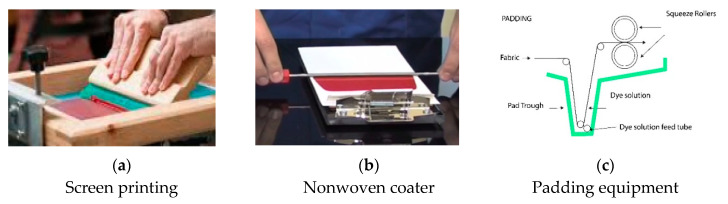
Description of chemicals used for experimental work.

**Figure 2 sensors-20-06816-f002:**
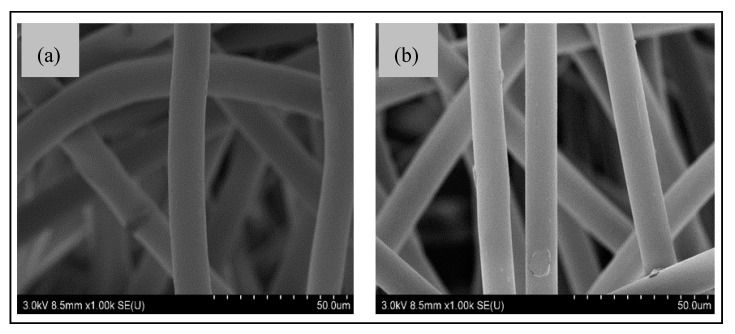
Scanning electron microscope (SEM) images of (**a**) untreated PET, (**b**) atmospheric plasma (ATMP)-treated fiber surface of nonwovens.

**Figure 3 sensors-20-06816-f003:**
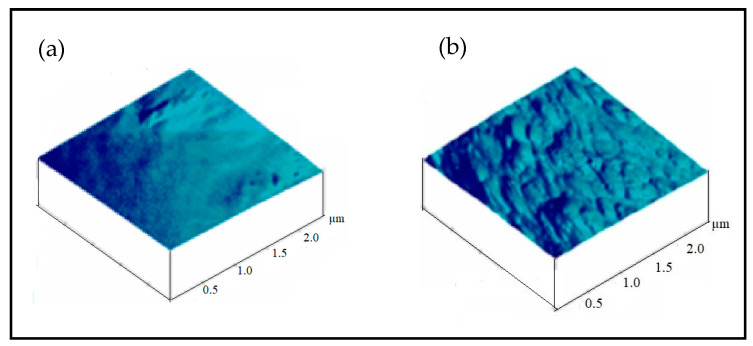
Atomic force microscopy (AFM) images of (**a**) untreated PET, (**b**) ATMP-treated fiber surface of nonwovens.

**Figure 4 sensors-20-06816-f004:**
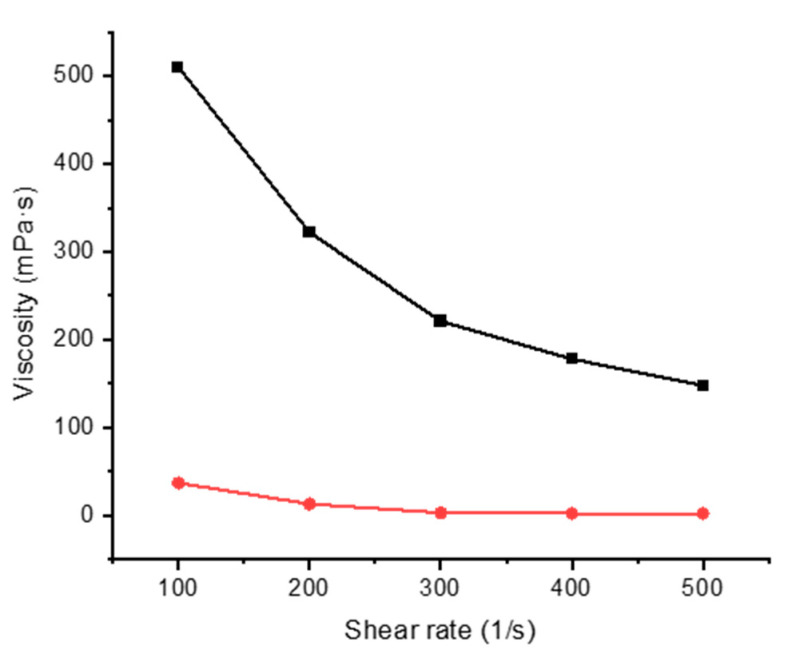
Viscosity of 5% gelatin with varying shear rate at 20 °C (black line) and 60 °C (red line).

**Figure 5 sensors-20-06816-f005:**
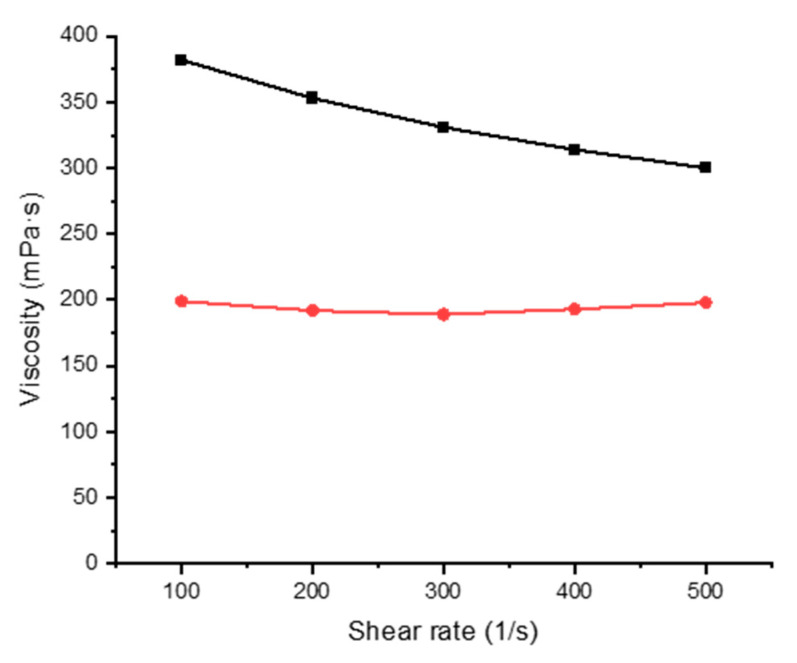
Viscosity of 5% sodium alginate with the varying shear rate at 20 °C (black line) and 60 °C (red line).

**Figure 6 sensors-20-06816-f006:**
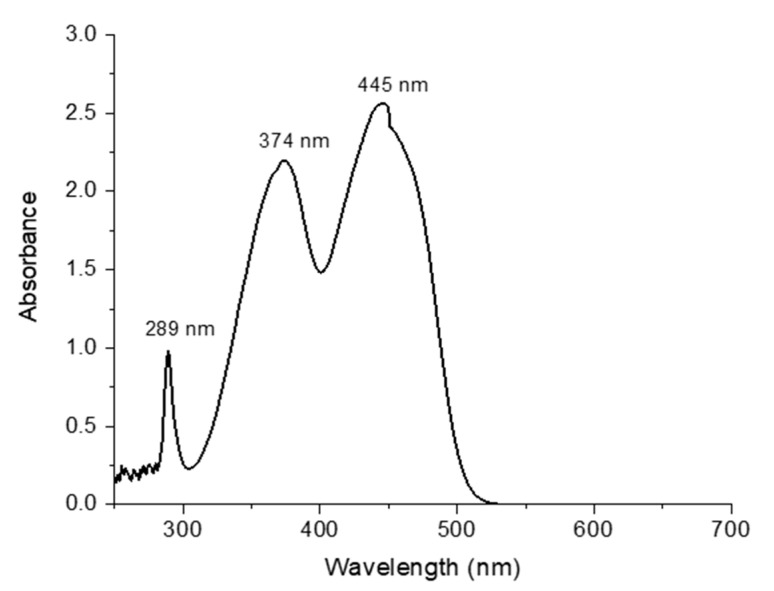
UV–visible spectroscopy of FMN solution.

**Figure 7 sensors-20-06816-f007:**
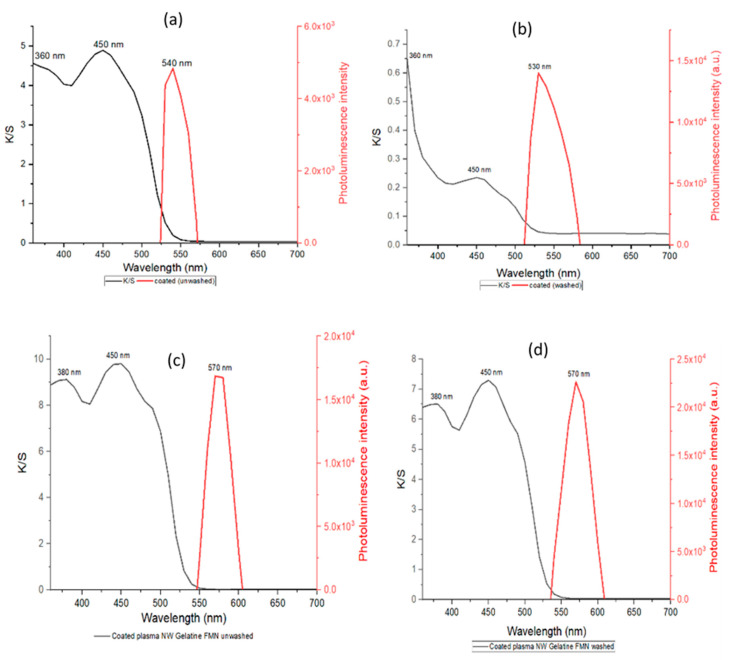
Absorbance of FMN gelatin mixture coated on (**a**) untreated PET nonwoven coated, unwashed sample (**b**) untreated PET nonwoven coated, washed sample (**c**) plasma-treated PET nonwoven coated, unwashed sample (**d**) plasma-treated PET nonwoven coated, washed sample.

**Table 1 sensors-20-06816-t001:** Description of chemicals used for experimental work.

Description	Flavin Mononucleotide (FMN)	Gelatin	Sodium Alginate (Alginate)
Structure	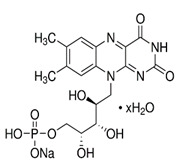	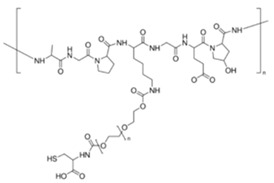	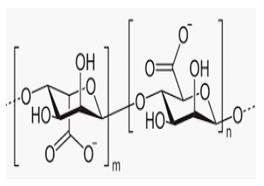
Water solubility	100 g/L at 20 °C	50 mg/mL, H_2_O	soluble
Molecular/Formula weight	478.33 g/mol	% Protein = 77	216.12 g/mol
Polymer weight	-	Gel strength (bloom no.) = 300	-
Viscosity	-	7.89 cps (6.67%, Water @ 33 °C [21])	24.4 cps (1%, Water @ 25 °C)

**Table 2 sensors-20-06816-t002:** Details of biopolymer film preparation and conditions.

Biopolymer (5%)	Active Agent (1%, 5%)	Max Temp	(H_2_O) Volume
Gelatin	FMN	60 °C	100 mL
Alginate		80 °C	100 mL

**Table 3 sensors-20-06816-t003:** Water contact angle and capillary uptake.

Sample Description	Water Contact Angle	Capillary Uptake
PET nonwoven (untreated) 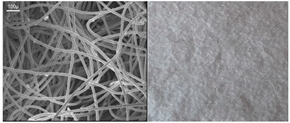	141°	0 mg
PET nonwoven treated by air atmospheric plasma treatment (60 kJ/m^2^) 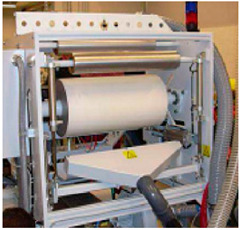	0°	1460 mg

**Table 4 sensors-20-06816-t004:** Biopolymer film images under ultraviolet (UV) light and daylight (**a**) 5% alginate +1 mL of 5% FMN (**b**) 5% alginate +5 mL of 5% FMN (**c**) 5% gelatin +1mL of 5% FMN (**d**) 5% gelatin +5 mL of 5% FMN.

Sample (a)	Sample (b)	Sample (c)	Sample (d)
5% alginate + 1 mL of 5%FMN	5% alginate + 5 mL of 5%FMN	5% gelatin + 1 mL of 5%FMN	5% gelatin + 5 mL of 5%FMN
Under daylight			
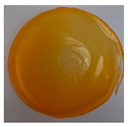	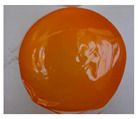	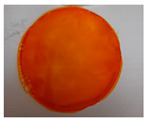	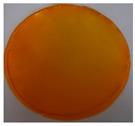
Under UV light			
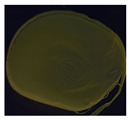	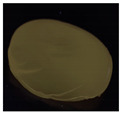	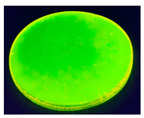	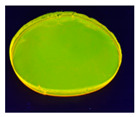

**Table 5 sensors-20-06816-t005:** Images of fabric under daylight and UV light.

Fabric	Undyed Fabric under Daylight	Dyed Fabric under Daylight	Dyed Fabric under UV Light
Cotton	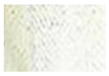		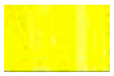
PET-polyester	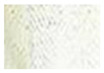		
Viscose	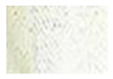		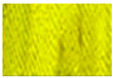

**Table 6 sensors-20-06816-t006:** Images of biopolymer film and coated nonwoven samples under daylight and UV light (370 nm). (a) gelatin film without FMN, (b) gelatin film with FMN under daylight, and (c) gelatin film with FMN under UV light, (d) untreated PET (e) PET nonwoven after coating with gelatin with FMN under daylight, (f) PET nonwoven after coating with gelatin with FMN under UV light.

	Under Daylight	Under UV Light
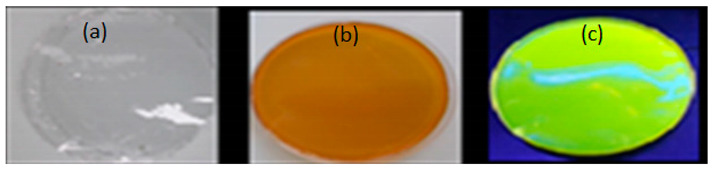		
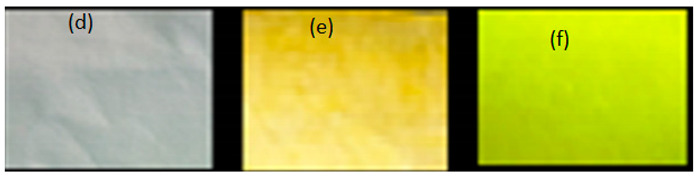		

**Table 7 sensors-20-06816-t007:** Screen printing on nonwoven PET fabric.

Screen Printed		Nonwoven Untreated	Nonwoven Plasma Treated
Under UV Light			
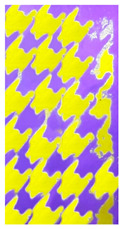	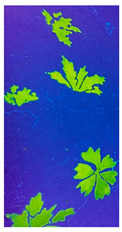	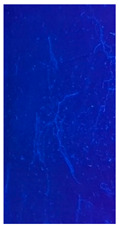	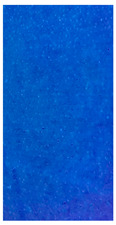

**Table 8 sensors-20-06816-t008:** Color strength (K/S) values and photoluminescence intensity values at the maximum emission wavelength (570 nm).

PET nonwoven	K/S Value (364 nm)	K/S Value (470 nm)	Intensity Observed at 570 nm for Excitation Wavelength (364 nm) Intensity Observed at 443 nm for Excitation Wavelength (364 nm)	Intensity Observed at 570 nm for Excitation Wavelength (470 nm) Intensity Observed at 564 nm for Excitation Wavelength (470 nm)
plasma treated			16,000	7000
plasma treated PET coated with gelatin-FMN	9	10	23,600	16,500
plasma treated PET coated with gelatin-FMN (washed sample)	6.5	7	28,900	30,000
plasma treated PET padded with acrylate-FMN	14	15	15,900	17,000
plasma treated PET padded with acrylate-FMN (washed sample)	10	12	15,600	21,000

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
