# Peer review of "Glow-in-the-Dark Patterned PET Nonwoven Using Air-Atmospheric Plasma Treatment and Vitamin B2-Derivative (FMN)"

_sensors, 2020, doi:10.3390/s20236816_

Round 1

Reviewer 1 Report

In this manuscript, the authors report a method for generating fluorescent patterned materials using a vitamin B2 derivative (FMN) based on surface plasma treatment of hydrophobic surface of PET nonwoven fibers.  These patterned materials can be potentially useful as light-emitting textiles. The experiments were well designed and the results were well presented. I strongly recommend for publication in Sensors after some minor corrections.

  1. In the experimental section on page 3, it is necessary to show how long the PET fibers were treated under plasma.
  2. Please explain why only the films with gelatin showed yellow-green fluorescence not the alginate films.

Reviewer 2 Report

The manuscript describes different methods to prepare photoluminescent non woven PET textiles with flavin mononucleotide (FMN) derived from vitamin B2. The FMN mixed with gelatin was screen printed; coated or padded on PET non wovens. The results showed that to have an homogeneous and durable treatment the non woven PET needed an air atmospheric plasma pre-treatment. 

The research is interesting and the experimental work is complete. The results discussion can be improved:

  • Table 5: the author reports the results of a previous work. The methods to dye cellulose fibers and PET are not described in the present work, only as a reference. The author should decide if to describe the methods previously used and show the results or just add the reference.
  • are the non woven PET reported in table 6 coated as reported in line 266-268? 
  • the work showed that the plasma pre-treatment on PET improved the durability of the treatment. The author should reports the washing conditions and the number of washing cycles
  • line 309. Discussions and conclusions. The paragraph needs to be improved. There are many information that describe the methods used to prepare the films on petri dishes and the coating of non woven PET. Biopolymers films preparation and compositions are described as well as PET treatment but the details and information are mixed togheter so as it is confusing. 
